# Biased Opioid Receptor Agonists: Balancing Analgesic Efficacy and Side-Effect Profiles

**DOI:** 10.3390/ijms26051862

**Published:** 2025-02-21

**Authors:** Jie Ju, Zheng Li, Jie Liu, Xiaoling Peng, Feng Gao

**Affiliations:** Department of Anesthesiology and Pain Medicine, Hubei Key Laboratory of Geriatric Anesthesia and Perioperative Brain Health, Wuhan Clinical Research Center for Geriatric Anesthesia, Tongji Hospital, Tongji Medical College, Huazhong University of Science and Technology, Wuhan 430030, China; m202276266@hust.edu.cn (J.J.); d202282261@hust.edu.cn (Z.L.); d202382277@hust.edu.cn (J.L.); d202282262@hust.edu.cn (X.P.)

**Keywords:** opioids, biased agonist, β-arrestin, TRV130, PZM21

## Abstract

Opioids are the most effective option for severe pain. However, it is well documented that the side effects associated with prolonged opioid use significantly constrain dosage in the clinical setting. Recently, researchers have concentrated on the development of biased opioid receptor agonists that preferentially activate the G protein signaling pathway over β-arrestin signaling. This approach is based on the hypothesis that G protein signaling mediates analgesic effects, whereas β-arrestin signaling is implicated in adverse side effects. Although certain studies have demonstrated that the absence or inhibition of β-arrestin signaling can mitigate the incidence of side effects, recent research appears to challenge these earlier findings. In-depth investigations into biased signal transduction of opioid receptor agonists have been conducted, potentially offering novel insights for the development of biased opioid receptors. Consequently, this review elucidates the contradictory roles of β-arrestin signaling in the adverse reactions associated with opioid receptor activation. Furthermore, a comparative analysis was conducted to evaluate the efficacy of the classic G protein-biased agonists, TRV130 and PZM21, relative to the traditional non-biased agonist morphine. This review aims to inform the development of novel analgesic drugs that can optimize therapeutic efficacy and safety, while minimizing adverse reactions to the greatest extent possible.

## 1. Introduction

Opioids, such as morphine, fentanyl, and hydrocodone, are extensively utilized in clinical practice as the mainstay for managing various types of moderate to severe pain [1]. Despite potent analgesic properties, the extensive range of side effects greatly limits the use of opioid drugs, such as nausea and vomiting, constipation, respiratory depression, addiction, tolerance, opioid-induced hyperalgesia (OIH), etc. [2,3]. Even the Enhanced Recovery After Surgery (ERAS) protocols have increasingly advocated for the reduction or elimination of opioids, aiming for opioid-free analgesia [4]. However, opioids continue to be the most effective treatment for moderate to severe pain [4,5]. Therefore, the limitations inherent to conventional opioid drugs present significant challenges in clinical management. The addictive properties and euphoric effects of opioids threaten a significant market share in the illicit drug trade [6]. Furthermore, the mortality rate associated with opioids has remained consistently elevated in recent years, imposing considerable burdens on families and society [7,8,9]. These factors have made the development of analgesics targeting opioid receptors a challenging yet essential endeavor. Consequently, the pursuit of more potent analgesics with improved efficacy and minimal side effects has become a focused objective.

To mitigate the severe side effects caused by opioid receptor agonists as much as possible, researchers have proposed various strategies. Personalized administration is tailored to individual patients, using sustained-release formulations to reduce overall dosage and combined with low-dose antagonists to reduce adverse reactions [10,11]. Multiple opioid receptor agonists can also be used to counteract the side effects caused by one subtype, while preserving positive therapeutic efficacy through simultaneous activation of multiple opioid receptors [10]. Designing opioid receptor subtype-selective agonists and positive allosteric modulators has not effectively mitigated multiple side effects [12,13]. The advancement of selective agonists has consistently been a focal point for researchers. G protein-selective agonists, which are engineered to specifically interact with opioid receptors and specific second messenger systems, have demonstrated a substantial reduction in the incidence of adverse opioid reactions, including compounds such as TRV130 and PZM21 [14,15] (Table 1). Notably, TRV130 has received formal approval for clinical use.

In recent years, novel insights have emerged into the MOR-mediated biased agonistic signaling pathway, providing robust support for the prospective development of more precise and efficacious biased agonists. This review synthesizes the mechanisms underlying G protein-biased opioid receptor agonists and evaluates the therapeutic efficacy of classical biased opioid receptor agonists in comparison to the non-biased agonist morphine in both preclinical research and clinical practice. This comparative analysis establishes a foundational framework for the advancement of analgesics that are safer and have fewer side effects.

## 2. Mechanism of Opioid-Induced Analgesia and Adverse Reactions

Opioid receptors, which are G protein-coupled receptors (GPCRs), are primarily categorized into four subtypes: mu (μ)-opioid receptor (MOR), delta (δ)-opioid receptor (DOR), kappa (k)-opioid receptor (KOR), and nociceptive nocicepin/orphanin FQ peptide receptors (NOPR). All opioid receptor subtypes have analgesic implications. The analgesic mechanisms of opioids have been the subject of extensive investigation. Most clinically utilized opioids exert analgesic effects through interaction with MOR, an inhibitory G protein-coupled receptor [14,28] that is abundantly expressed in the central nervous system and gastrointestinal tract [29,30,31,32].

Upon binding to MOR, exogenous opioids induce conformational changes in the receptor, thereby modulating cellular activities through G protein and β-arrestin signaling pathways [33,34] (Figure 1). MOR is activated and coupled with G protein, which subsequently dissociates into alpha (Gα) and beta-gamma (Gβ/γ) subunits. Gα inhibits the activity of adenylate cyclase (AC), thereby preventing adenosine triphosphate (ATP) from producing cyclic adenosine monophosphate (cAMP) [35]. Gβ/γ leads to the inactivation of calcium channels (reducing Ca^2+^ influx) and activates potassium channels (increasing K^+^ efflux) [36,37]. Consequently, these effects inhibit the release of nociceptive neurotransmitters, thereby reducing the transmission of pain signals and exerting analgesic effects. In contrast, MOR is phosphorylated by G protein-coupled receptor kinase (GRK), which recruits β-arrestin. The binding of β-arrestin hinders the interaction between MOR and G protein, thereby blocking G protein-dependent signaling. Additionally, β-arrestin recruitment leads to endocytosis and desensitization of MOR and activation of the mitogen-activated protein kinase (MAPK) signaling pathway [34,38,39]. These processes are implicated in various adverse effects [40,41,42,43,44,45]. Empirical evidence has also confirmed these findings; for instance, in mice genetically deficient in β-arrestin, morphine administration has been associated with a reduction in constipation [46], respiratory suppression [47,48], and tolerance [14,47], among others.

## 3. Conflicts in the β-Arrestin Signaling Pathway

The adverse effects of opioids can be reversed by MOR antagonists and are absent in animals with MOR gene knockout [29,30,49], suggesting that the incidence of side effects may not be associated with receptor selectivity. Previous studies have established that the analgesic effect of opioid receptor agonism is mediated by G protein signaling, whereas side effects such as respiratory depression, gastrointestinal dysfunction, and tolerance are mediated by β-arrestin signaling. However, a recent study demonstrated that both G protein signaling and β-arrestin contribute to side effects induced by morphine [50]. This study found that the β-arrestin pathway is involved in morphine tolerance, whereas the G protein signaling pathway is implicated in respiratory depression and constipation, achieved through the design of permeable peptides in combination with morphine. This challenges existing theories. Initially, β-arrestin was regarded as a negative regulator of the G protein signaling pathway. However, it was later discovered that β-arrestin can also function as a signal transducer of GPCR, mediating the transmission of downstream signals independently of G protein [51]. Indeed, the role of the β-arrestin signaling pathway in mediating the side effects of opioids remains unclear, and numerous studies investigating the adverse effects of opioids have focused on β-arrestin-2, although the findings remain contentious.

Early research in rodents revealed that the absence or reduction in the β-arrestin-2 gene amplifies the anti-nociceptive effects while simultaneously reducing tolerance to opioids [14,23,47,48,52,53,54,55], implying a role for β-arrestin-2 in the mechanisms underlying opioid-induced antinociceptive tolerance. Nonetheless, methadone, an opioid drug, exhibits a high affinity for β-arrestin-2, yet results in less tolerance and dependency than morphine [56]. This phenomenon blurs the role of β-arrestin-2 in analgesic tolerance. Subsequently, He and colleagues administered various G protein-biased and non-biased agonists to WT mice, revealing an inverse correlation between β-arrestin-2 recruitment and analgesic tolerance. Additionally, mice with the RMOR (Recycling MOR) genotype exhibit resistance to analgesic tolerance [57]. Contrary to previous findings, drugs that effectively recruit β-arrestin-2 reduce analgesic tolerance.

Opioid-induced analgesic tolerance is commonly regarded as centrally mediated; however, nociceptive dorsal root ganglia also play a significant role in tolerance development [58,59]. Research has demonstrated that desensitization of μ-Opioid receptors mediated by β-arrestin-2 is implicated in the acute analgesic tolerance of dorsal root ganglion nociceptive neurons to opioids [60]. Notably, the absence of β-arrestin-2 does not affect long-term tolerance to morphine in either male or female mice [60], indicating that non-β-arrestin-biased agonists may not alleviate tolerance induced by prolonged opioid use. The experimental data from various laboratories exhibit significant variability, which may be attributed to factors such as drug dosage, dosing frequency, physiological condition of animals, and laboratory settings. These divergent results could profoundly impact the clinical application of biased agonists, necessitating more rigorous experiments to obtain conclusive evidence.

Respiratory depression is another controversial side effect mediated by β-arrestin signaling. The risk of respiratory depression constrains the clinical dosage of opioids [61]. Numerous studies have demonstrated that the deletion of the β-arrestin-2 gene exhibits less morphine-induced respiratory depression [46,47,48,55,62], indicating a mediating role of β-arrestin-2 in respiratory dysfunction. However, previous perspectives have posited that opioid-induced respiratory depression is related to G protein-gated inwardly rectifying potassium channels rather than β-arrestin signaling [63,64,65]. Recent studies provide additional support for this perspective. Kliewer et al. proposed that morphine-induced respiratory depression is not associated with β-arrestin-2 signaling [66]. Afterward, He et al. investigated the drug response of three different genotypes—wild type (WT), β-arrestin-2 knockout (KO), and Recycling MOR (RMOR) C57BL/6 mice—to morphine. Their findings indicated that all three genotypes exhibited comparable levels of respiratory inhibition and analgesic effects in response to morphine [57]. Furthermore, when WT mice were administered different G protein-biased and non-biased agonists, no significant difference was observed in the maximum respiratory inhibition. These findings suggest that there may be no significant correlation between the recruitment level of β-arrestin-2 and the extent of respiratory depression.

Studies have shown that biased MOR agonists have the potential to mitigate abuse relative to standard opioids [67]. Notably, β-arrestin signaling may not be implicated in the addiction mechanism associated with opioid receptor agonists. Research has shown that in mice with β-arrestin-2 gene knockout, morphine-induced striatal dopamine release is significantly increased, and conditioned place preference (CPP) behavior is markedly enhanced [46,48,68]. This suggests that opioid receptor agonists that preferentially activate G protein over β-arrestin signaling may possess the potential to induce physical dependence [48].

## 4. What Is the Biased Ligand Agonist?

Previously, it was believed that ligands exert different pharmacological effects due to their interaction with distinct receptors [69]. However, it has gradually become evident that the effects of ligands are not entirely dependent on receptor selection; rather, they are closely associated with signaling molecules downstream of the receptors [70]. The activated conformation of GPCRs is dynamically variable, which allows for the recruitment of G proteins and β-arrestin to varying extents, thereby triggering different signaling pathways [71]. Recent studies have shown that ligands do not uniformly activate signaling pathways downstream of GPCRs; instead, they can selectively and preferentially activate certain pathways, a phenomenon known as “biased activation” [72]. Currently, there is no universally accepted standard for evaluating and assessing ligand biases. Some researchers have argued that ligand activation-specific signals are influenced by factors such as ligand bias, system bias, and dynamic bias [73,74,75]. The concept of the “bias factor” has been introduced as a measure of the selectivity of MOR agonists [73,76]. This involves systematically testing the degree of bias among different MOR agonists (i.e., the potency differences between G protein signals and other signaling pathways) to determine the therapeutic index [77].

Biased agonists stabilize receptor conformations and preferentially activate specific signaling pathways, allowing for more precise regulation of cellular functions. However, significant challenges persist in the development of biased agonists [78]. There is a pressing need for further technical advancements in the quantitative analysis of bias and the establishment of appropriate screening methods. Additionally, identifying which cell types express the drug targets, elucidating the signaling pathways that can elicit therapeutic effects, and understanding the regulatory mechanisms involved are all critical areas that require thorough investigation.

## 5. A New Approach to Biased Agonists

When non-biased agonists activate MOR, G protein and β-arrestin compete for the same binding site, which forms strong polar interactions with ICL2 (intracellular loop 2 (ICL2)) and ICL3 or the cytoplasmic region of TM6, and facilitates coupling with MOR [79]. Recent structural biology and pharmacological experiments have revealed that opioid receptor agonists attenuate or abolish the β-arrestin signaling pathway, resulting in a pattern of G protein-biased signal transduction. This bias reduces the interaction between opioids and the sixth and seventh transmembrane regions of the MOR [80]. Further verification was achieved through cellular-level functional analysis and molecular dynamics simulations. Specifically, studies have demonstrated that G protein-biased agonists (such as TRV130, PZM21, and SR-17018) preferentially bind to the TM2/3 region of the MOR binding site. This interaction induces the repositioning of TM6 in the cytoplasmic region of MOR, thereby preventing β-arrestin from forming a polar anchor with MOR, resulting in decreased affinity [79,80]. In contrast, balanced agonists such as fentanyl engage in more extensive and balanced interactions with the transmembrane region of the MOR [80]. Based on this, researchers meticulously modified fentanyl and synthesized two derivatives, FBD1 and FBD3, which exhibited a preference for the G protein pathway, thereby further substantiating the concept of biased binding sites. This discovery provides novel insights for further design of biased opioids. Nonetheless, the molecular binding model derived from molecular docking and dynamic simulation is intricate and may not fully replicate the actual drug binding mode. It is anticipated that this groundbreaking discovery will require further validation in animal models.

## 6. Comparison Between Biased Agonists and Morphine

MOR is the principal target of opioids, such as morphine and hydrocodone. Numerous pharmacological studies have focused on the ligand bias of this receptor. Some studies have measured the bias coefficient of MOR agonists at the molecular level as well as the analgesic and side effects at the animal level. These results indicate that when a biased ligand activates the G protein-dependent signaling pathway, the safety window is broad [21]. Consequently, biased ligand theory posits that biased agonists of the G protein-dependent signaling pathway may produce more selective pharmacological profiles and physiological responses. Based on this theory, extensive screening of compounds was conducted, resulting in the identification of a series of potentially biased ligands such as TRV-130 and PZM21.

### 6.1. TRV130

In 2013, TRV130 was designated as Oliceridine or OLINVO, demonstrating a selectivity for MOR receptors that exceeds 400-fold that of other opioid receptors [14]. Structurally distinct from other MOR agonists, TRV130 exhibits a degree of G protein coupling activation comparable to morphine; however, its efficiency in recruiting β-arrestin-2 is approximately 14% of that observed with morphine [14,81]. TRV130 possesses unique pharmacokinetic properties characterized by a rapid onset of action and prolonged duration. Clinical studies have indicated that its effects commence within 2-5 min and persist for up to 3 h [81]. The therapeutic window for dosage is broad, obviating the necessity for dosage adjustments across diverse patients, including variations in age, sex, race, body weight, and hepatic or renal function [82]. Consequently, TRV130 is regarded as a highly efficacious, safe, and selective drug characterized by its G protein-biased MOR agonist properties.

Similarly to morphine, TRV130 exhibits a high volume of distribution and significant monophasic clearance [14]. Its lipophilic properties facilitate rapid penetration into the brain, resulting in widespread tissue distribution and prompt access to the central nervous system where it binds to opioid receptors to exert its pharmacological effects [14]. Numerous animal studies have evaluated the efficacy of TRV130 in comparison with morphine, intending to assess the relative advantages and disadvantages of each, thereby improving clinical practices (Table 2). Empirical studies have demonstrated that TRV130 induces a more pronounced analgesic effect than respiratory depression and sedation in rat models. Furthermore, TRV130 has been shown to produce potent analgesic efficacy, with its effectiveness in acute thermal injury models in rats and mice ranging from 4.5 to 10 times greater than that of morphine [14,52,83,84]. However, at equivalent analgesic doses, TRV130 exhibits a less pronounced sedative effect than morphine, which aligns with its reduced inhibition of the central nervous system [14]. Additionally, the weak activation of β-arrestin-2 by TRV130 can lead to respiratory depression, constipation, antinociceptive tolerance, and behaviors associated with abuse [52,77,83,85,86], similar to other MOR agonists. However, multiple clinical trials have demonstrated that TRV130 elicits more rapid antinociceptive effects than morphine at equivalent analgesic doses in patients with moderate to severe pain. Furthermore, TRV130 is associated with a reduced incidence of adverse events, including nausea, vomiting, respiratory dysfunction, and gastrointestinal disturbances, thereby offering a broader therapeutic window [14,15,87,88,89,90,91]. Additionally, various rodent studies have indicated that TRV130 shows a lower propensity for tolerance development than conventional opioid medications [83,84,92,93]. Cellular experiments further substantiated that, compared to morphine, TRV130 exhibited reduced receptor phosphorylation and internalization, processes linked to opioid tolerance, when co-cultured with human MOR-expressing cells [14]. Interestingly, TRV130 has been observed to induce opioid-induced hyperalgesia (OIH), albeit to a lesser extent than morphine. In contrast, repeated or continuous administration of TRV-0109101, an analog of TRV130, does not result in OIH; however, it does lead to significant tolerance, with a rate of drug resistance comparable to that observed with conventional mu-opioid receptor (MOR) agonists [55,94].

Conversely, empirical evidence suggests that TRV130 exhibits abuse potential comparable to that of hydrocodone [77]. Within the rat intracranial self-stimulation (ICSS) paradigm, TRV130 demonstrated reward-enhancing effects analogous to those of conventional opioids [83]. Notably, a single analgesic dose of TRV130 did not induce conditioned place preference (CPP) in mouse models [52]. Furthermore, at equianalgesic doses, TRV130 elicited a lower CPP response than morphine; however, this disparity was mitigated when higher doses of TRV130 were administered [84]. This phenomenon may be attributed to the regulation of brain exposure to opioid drugs by the P-glycoprotein (P-gp) efflux transporter (MDR1) [96,97,98,99,100,101]. The function of P-gp is a critical determinant of the pharmacokinetic properties of opioids. Both TRV130 and morphine are substrates of P-gp; however, administering morphine at doses at least ten times higher than TRV130 may saturate efflux transporters, thereby reducing morphine clearance and resulting in increased brain uptake. This suggests that the dosage required for TRV130 to achieve analgesic effects may not pose a significant risk of abuse. Moreover, TRV130 has demonstrated opioid withdrawal mitigation effects comparable to those of methadone [102], indicating its potential as a viable treatment option for patients with opioid use disorder (OUD). Empirical studies have revealed that TRV130 exerts a more pronounced impact on drug addiction relapse in male rats than in female rats [85], suggesting that maintenance therapy with TRV130 may generate greater efficacy in male patients. Furthermore, prolonged administration of TRV130 has been shown to prevent brain hypoxia induced by moderate doses of hydrocodone in both male and female rats [103], with this neuroprotective effect sustained across both sexes.

Morphine-6-glucuronide, the active metabolite of morphine, causes long-term respiratory inhibition [104,105]. In contrast, TRV130 lacks bioactive metabolites [106]. The high potency and rapid pharmacokinetics of TRV130 may contribute to a safer profile for pain management. Owing to its pharmacokinetic properties, TRV130 is limited to intravenous administration in humans [87]. In 2020, the U.S. Food and Drug Administration (FDA) approved TRV130 for short-term intravenous use in hospital settings, with a daily cumulative dose not exceeding 27 mg. The limitation associated with the unsuitability for daily maintenance can be mitigated through oral administration of the active analog TRV734 [17,107] or by formulating depot preparations of TRV130/TRV734 for periodic application. Additionally, a method for chronic pain management via transdermal patch administration, leveraging its high skin permeability, is currently under development.

The biased agonism of TRV130 has been shown to offer significant advantages in mitigating side effects relative to other MOR agonists. An economic model indicated that the administration of TRV130 in postoperative care for high-risk patients, including elderly and obese individuals who are susceptible to opioid-related adverse events (ORAEs), yields significant health and economic benefits [108]. Furthermore, preclinical studies have shown that TRV130 is well tolerated in animal models and exhibits superior therapeutic efficacy relative to morphine. However, numerous clinical studies conducted since 2014 have indicated the persistence of opioid-related side effects. Therefore, there is a pressing need for a more comprehensive investigation of biased agonists in the clinical setting.

### 6.2. PZM21

Unlike TRV130, the design strategy of PZM21 is based on analysis of the solved MOR structure and is further developed through high-throughput screening [52]. It exhibits high selectivity for the MOR and is associated with weak KOR activation [52]. Similarly to TRV130, PZM21 activates the G protein and reduces the recruitment of β-arrestin [52,109]. The respiratory effects of PZM21 differ significantly from those of morphine. Initial studies indicated that mice administered PZM21 do not experience respiratory depression, whereas equivalent doses of morphine result in respiratory inhibition [52]. However, recent research has indicated that PZM21 exhibits respiratory inhibitory effects comparable to morphine and induces antinociceptive tolerance [110]. These findings may be attributed to variations in experimental outcomes across different laboratories. Furthermore, studies in mice have demonstrated that PZM21 selectively inhibits the affective components of pain. Notably, PZM21 produced analgesic effects in the hot plate test, which assesses the advanced central nervous system and spinal cord injury circuits, but did not provide pain relief in the tail-flick test, which evaluates spinal reflexes [52]. This presents a novel distinction between PZM21 and other opioids. Through simulation, it was determined that PZM21 did not exhibit significant advantages in terms of analgesia or reduction in side effects, such as TRV130 [16,110]. Specifically, PZM21 was identified as a low-efficacy analgesic and a low-efficacy biased agonist. These findings are consistent with the experimental results reported by Hill et al., which demonstrated PZM21’s inefficacy in both the G protein and β-arrestin pathways [110]. Unlike TRV130, this study assessed the abuse potential of PZM21 using intracranial self-stimulation and found minimal enhancement of its use. These results suggest the need for further investigation of the potential applications and limitations of PZM21.

## 7. Other Biased Agonists

SR-17018 has a half-life of approximately 6 h, demonstrates brain permeability, and is associated with low respiratory depression [21,22]. Long-term oral administration of SR-17018 in mice produces analgesic effects comparable to those of morphine. Upon cessation of treatment, withdrawal symptoms were observed, albeit for a shorter duration than those associated with morphine. When administered to male mice at a dose of 24 mg/kg/day, SR-17018 did not induce drug resistance; however, a slight reduction in efficacy was noted at a dose of 48 mg/kg/day. These findings suggest that the tolerance to SR-17018 may be dose-dependent. The absence of tolerance in female mice administered SR-17018 at a dose of 48 mg/kg/day indicates the potential sex-specific effects of SR-17018 [23]. However, SR-17018 is more effective in various pain models and does not induce tolerance, which distinguishes it from other commonly used opioids [111]. Prolonged administration of SR-17018 preserves sensitivity to morphine in mice and mitigates morphine withdrawal symptoms, likely due to SR-17018’s role in stabilizing the G protein-coupled state. Thus, SR-17018 may be a viable strategy for restoring MOR responsiveness and sustaining opioid analgesic efficacy.

SR-14968 is a complete agonist with less respiratory depression and a broader therapeutic window [21]. SR-14968 exhibits a G protein bias approximately ten times greater than TRV130 [14,21]. This compound demonstrated dose-dependent analgesic and differential stimulatory effects. However, compared to morphine and methadone, SR-14968 showed a higher propensity to produce discriminative stimuli. The extended duration of action of SR-14968, relative to TRV130, morphine, and methadone in rodent models, raises questions about how pharmacokinetic properties influence enhancing effects [21]. Furthermore, no sex differences were observed in the efficacy of SR-14968 [95]. In addition, SR-17018 and SR-14968 are both non-competitive agonists that can stabilize the receptor active. Studies have demonstrated that SR-17018 and SR-14968 bind to MOR in an almost irreversible manner, resulting in sustained G protein signaling. However, MOR antagonists can completely reverse this stimulation, indicating that SR series compounds bind to different sites on the receptor with high affinity, promote G protein signaling, and maintain sensitivity to orthosteric antagonists [112]. Furthermore, SR-17018 can inhibit the effects of SR-14968, suggesting that there may be competition between them at the allosteric site [112].

## 8. Conclusions

With the progressive rise in demand for opioids, the impact of side effects is becoming increasingly apparent. The development of targeted, safer, and more potent analgesics with a broader therapeutic window aligns with contemporary trends in pharmaceutical research and clinical practice. Although the existing designs of biased opioid receptor agonists have not yet fully met the rigorous standard of “zero side effects”, the ongoing advancement of biased ligand technology offers promise for the creation of innovative opioid therapeutics.

## Figures and Tables

**Figure 1 ijms-26-01862-f001:**
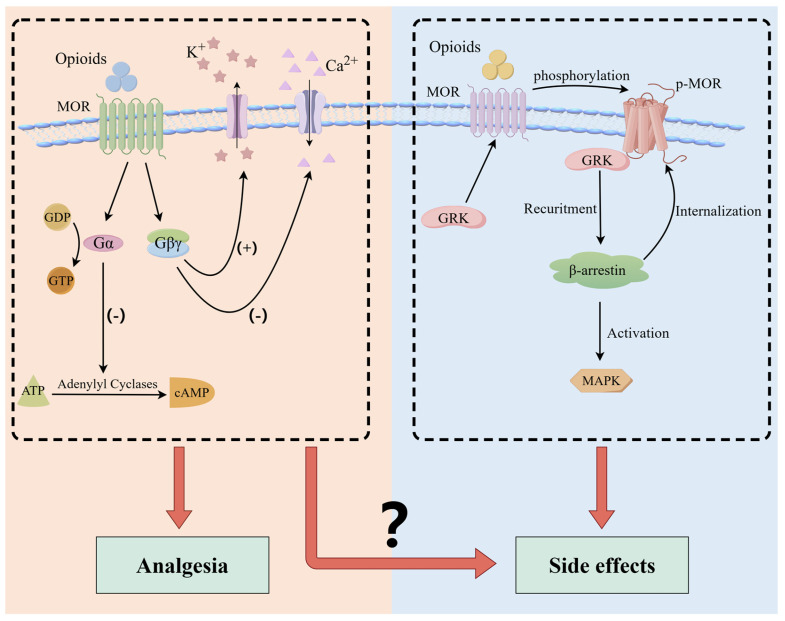
Intracellular signaling mediated by opioid receptors. Activation of opioid receptors by exogenous agonists leads to dissociation of G protein heterotrimers into α and βγ subunits. Gα inhibits adenylate cyclase activity, thereby preventing adenosine ATP from producing cAMP. Gβ/γ leads to the inactivation of calcium channels and activation of potassium channels. Consequently, these effects inhibit the release of nociceptive neurotransmitters, thereby reducing the transmission of pain signals and exerting analgesic effects. The interaction of β-arrestin with phosphorylated MOR results in receptor internalization and desensitization, and activates the MAPK cascade pathways.

**Table 1 ijms-26-01862-t001:** Examples of biased ligands for opioid receptors.

Ligand	Receptor	Signaling Bias	Effect	Reference
TRV130	μ-Opioid	G-protein	Antinociceptive, less tolerance, and respiratory depression.	[14]
PZM21	high selectivity for μ-Opioid with weak κ-Opioid	G-protein with weak β-arrestin	Antinociceptive, no respiratory depression, and gastrointestinal disorders.	[16]
TRV734	μ-Opioid	G-protein	Potently analgesic, less gastrointestinal dysfunction, well tolerated.	[17]
SHR9352	high selectivity for μ-Opioid over κ-opioid and δ-opioid	G-protein	Antinociceptive, almost no constipation, good central nervous system penetration, well tolerated.	[18]
Dmt-c[D-Lys-Phe-Asp]NH2	μ-Opioid and δ-Opioid	β-arrestin	Antinociceptive, decreased GI transit.	[19]
Mitragynine pseudoindoxyl	Mixed μ-Opioid and δ-Opioid	G-protein	Antinociceptive, no reward or aversion, diminished antinociceptive tolerance, ventilatory depression, and GI transit inhibition.	[20]
SR17018	μ-Opioid	G-protein	A high ED50 for ventilatory suppression, no antinociceptive tolerance, withdrawal dissipates quickly.	[21,22,23]
SR-11501	μ-Opioid	β-arrestin-2	Respiratory suppression at lower doses.	[21]
SR-14968	μ-Opioid	G-protein	Antinociceptive, fentanyl-like discriminative stimulus effects, an improved potency ratio (drug discrimination potency/tail-flick potency).	[21]
SR-15098	μ-Opioid	G-protein	Antinociceptive, less respiratory depression.	[21]
SR-15099	μ-Opioid	G-protein	Antinociceptive, less respiratory depression.	[21]
MEL-N1606	μ-Opioid	G-protein	Potently analgesic, less tolerance and constipation, good central nervous system penetration, low addiction.	[24]
N-(3-fluoro-1-phenethylpiperidine-4-yl)-N-phenyl propionamide (NFEPP)	μ-Opioid	_	Only binds to MOR under acidic conditions and selectively activates MOR at damaged sites; no respiratory depression, no constipation, no addiction.	[25,26]
LY03014	μ-Opioid	G-protein	Separation of analgesic effect and respiratory inhibition; less respiratory depression, less constipation, less tolerance; no hepatotoxicity.	[27]

**Table 2 ijms-26-01862-t002:** Comparison of adverse reactions between TRV130 and morphine.

Animal Models	Experimental Animal	Dosage (mg/kg)	Frequency	Administration	Side Effects (Compared to Morphine)	Reference
right tibial fracture followed by intramedullary pinning (postsurgical/posttraumatic pain model)	Male C57BL/6J mice (10–12 weeks of age)	Morphine: 20 mg/kg and 40 mg/kg; TRV130: 5 mg/kg and 10 mg/kg.	twice per day for four days	Subcutaneous injection	Physical dependence caused by morphine and TRV130 is similar; TRV130 has almost no tolerance; the OIH level of TRV130 is lower than morphine.	[84]
		Morphine: 5 mg/kg TRV130: 1.25 or 10 mg/kg.	once daily for 3 days	Subcutaneous injection	Equal analgesic dose of TRV130 does not cause CPP.	
		Morphine: 10 mg/kg and 20 mg/kg; TRV130: 2.5 mg/kg.	twice daily for 7 days.	Subcutaneous injection	TRV130 does not significantly exacerbate allodynia or worsen gait disturbances after tibial fracture or worsen gait disorders; TRV130 produces lower upregulation of TLR4.	
——	Male C57BL/6J mice	Morphine: 6 mg/kg; TRV130: 1 mg/kg.	once	Subcutaneous injection	TRV130 reaches peak at 5 min, while morphine at 30 min; the duration of action of TRV130 and morphine is similar, about 90 min; TRV130 causes less constipation.	[14]
	Male Sprague Dawley rats	Morphine: 3.0 mg/kg; TRV130: 0.3 mg/kg.	once	Subcutaneous injection/intravenous injection	TRV130 has less sedative effect than morphine and less respiratory depression.	
——	Male Swiss Webster mice	Morphine: 50 mg/kg; TRV130: 10 mg/kg.	three times per day for 3 consecutive days	Subcutaneous injection	TRV130 did not show analgesic tolerance or gastrointestinal function inhibition; tolerance development in the ileum; similar to the abuse effect of morphine.	[83]
——	Male and female Sprague Dawley rats	Morphine: 3.2–32 mg/kg; TRV130: 0.1–10 mg/kg.	once	Subcutaneous injection	TRV130: significantly more potent to produce discriminative stimulus vs. antinociceptive effects; more favorable potency ratios and efficacy ratios.	[95]

Abbreviations: OIH, opioid-induced hyperalgesia; CPP, conditioned place preference.

## Data Availability

Not applicable.

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
