# Peer review of "Biased Opioid Receptor Agonists: Balancing Analgesic Efficacy and Side-Effect Profiles"

_ijms, 2025, doi:10.3390/ijms26051862_

Round 1
Reviewer 1 Report
Comments and Suggestions for Authors
The article “Biased Opioid Receptor Agonists: Balancing Analgesic Efficacy and Side Effect Profiles” by Ju et al. seeks to evaluate the contributions of G protein signaling biased agonism to improved side effect profiles of opioid analgesics. Additionally, they evaluate the contradictions associated with this hypothesis. The authors provide a discussion about the comparisons of TRV130 (Oliceridine) and morphine in vivo. However, there are significant deficiencies in citation throughout the review. Overall, the authors provide some discussion about biased agonists and the contradictions associated with the G protein signaling bias hypothesis, however, this discussion could be refined. Currently, the message the authors are trying to convey is disordered and could use restructuring.
Major Revisions:
- The second paragraph of the introduction which discusses potential strategies to mitigate side effects needs appropriate citations throughout. Currently none are included.
- The MOR signaling section of the manuscript should be rewritten for clarification. Currently, it seems to indicate that the inhibition of adenylyl cyclase results in the inactivation of calcium and activation of potassium channels, the later thought to be regulated by the beta/gamma subunit. Again citation and clarity is lacking.
- Figure 1 - The MOR (GPCR in the figure) is phosphorylated resulting in the recruitment of βarrestins, this is not indicated by the figure. Rather, the figure suggests that MOR activation results in the phosphorylation of other proteins, leading to the recruitment of βarrestin. Additionally, this figure seems to imply that all side effects of opioids are mediated by βarrestin which is lacking appropriate citations. This figure is fundamentally misleading.
- Table 1 contains various biased agonists that are not included in the text that the authors should briefly address. Additionally, for ligand SHR9352, it is quoted as “excellent MOR”. This statement should be adjusted to fit with the descriptions of the other agonists as either highly selective for MOR or high affinity for MOR as “excellent” doesn’t make sense in this context.
- The authors should note for SR-14968 that it exhibits less respiratory suppression and a wider therapeutic window and is a FULL agonist. SR-17018 has also been shown to be affective in various pain models, including neuropathic pain, unlike other conventionally used opiates which is not discussed by the authors (see Pantouli et al, 2021, Neuropharmacology). Additionally, the authors should note that SR-17018 and SR-14968 show unique pharmacology compared to other biased agonists as they act as noncompetitive MOR agonists (see Stahl et al, 2021, PNAS).
- Throughout the document, there is no clear discussion on what biased agonism is or how it is determined even though the review is a discussion about biased MOR agonists. The authors should include a brief discussion on how biased agonism is determined between two assays. This is a major deficiency.
- There are various grammatical errors throughout the manuscript that need correcting. Some of these are minor, but some make it hard to determine what the authors are trying to convey.
Minor Revisions:
- The statement: “Researchers indicate that agonists targeting the MOR and KOR subtypes of opioid receptors are effective in pain alleviation.” should be reworded. All subtypes of opioid receptors have implications in analgesia.
- In the PZM21 section, the authors state that PZM21 is an analog of TRV130. This statement should be corrected as PZM21 is not an analog of TRV130, rather it was identified through high throughput screening.
Comments on the Quality of English Language- There are various grammatical errors throughout the manuscript that need correcting. Some of these are minor, but some make it hard to determine what the authors are trying to convey.
Author Response
Thank you very much for taking the time to evaluate this paper and for your valuable comments and suggestions. We think these are excellent suggestions. We have made the corresponding modifications based on the suggestions, as follows:
Comments 1: The second paragraph of the introduction which discusses potential strategies to mitigate side effects needs appropriate citations throughout. Currently none are included.
Response 1: Thank you very much for your suggestion. We introduced the appropriate citations to the second paragraph of the introduction, which can be shown in the red font on lines 49-55.
Comments 2: The MOR signaling section of the manuscript should be rewritten for clarification. Currently, it seems to indicate that the inhibition of adenylyl cyclase results in the inactivation of calcium and activation of potassium channels, the later thought to be regulated by the beta/gamma subunit. Again citation and clarity is lacking.
Response 2: We adjusted the structure and content of the MOR signaling section and cited the corresponding references for clarification. The content is highlighted in red font on lines 77-91.
Comments 3: Figure 1 - The MOR (GPCR in the figure) is phosphorylated resulting in the recruitment of βarrestins, this is not indicated by the figure. Rather, the figure suggests that MOR activation results in the phosphorylation of other proteins, leading to the recruitment of βarrestin. Additionally, this figure seems to imply that all side effects of opioids are mediated by βarrestin which is lacking appropriate citations. This figure is fundamentally misleading.
Response 3: We apologize for the error in Figure 1. Specifically, MOR phosphorylation by G protein-coupled receptor kinase (GRK) leads to the recruitment of β-arrestin. Furthermore, the side effects of opioids are not entirely mediated by β-arrestin signaling. Recent studies have demonstrated that G protein signaling also contributes to morphine-induced side effects. Consequently, we have thoroughly revised the incorrect annotation in the figure, as well as the corresponding content in the manuscript, as indicated in Figure 1.
Comments 4: Table 1 contains various biased agonists that are not included in the text that the authors should briefly address. Additionally, for ligand SHR9352, it is quoted as “excellent MOR”. This statement should be adjusted to fit with the descriptions of the other agonists as either highly selective for MOR or high affinity for MOR as “excellent” doesn’t make sense in this context.
Response 4: These biased agonists do not appear in the text because there are few studies at present, which cannot be discussed in detail. Therefore, it is only listed in Table 1. Second, we modified the description of SHR9352, which should be described as “highly selective for MOR”. The specific modifications are in red font in Table 1.
Comments 5: The authors should note for SR-14968 that it exhibits less respiratory suppression and a wider therapeutic window and is a FULL agonist. SR-17018 has also been shown to be affective in various pain models, including neuropathic pain, unlike other conventionally used opiates which is not discussed by the authors (see Pantouli et al, 2021, Neuropharmacology). Additionally, the authors should note that SR-17018 and SR-14968 show unique pharmacology compared to other biased agonists as they act as noncompetitive MOR agonists (see Stahl et al, 2021, PNAS).
Response 5: Thank you very much for the references provided. I believe they enhance the content of the paper. We have carefully reviewed and further optimized the descriptions of SR-14968 and SR-17018. The specific modifications are in red on lines 356-357, 362-362, and 370-377.
Comments 6: Throughout the document, there is no clear discussion on what biased agonism is or how it is determined even though the review is a discussion about biased MOR agonists. The authors should include a brief discussion on how biased agonism is determined between two assays. This is a major deficiency.
Response 6: This is indeed a significant omission in the paper. We discussed what a biased agonist is and how to identify it. Please refer to the section in red font on lines 174-197 for supplementary information.
Comments 7: There are various grammatical errors throughout the manuscript that need correcting. Some of these are minor, but some make it hard to determine what the authors are trying to convey.
Response 7: We thoroughly reviewed the entire text, corrected grammatical errors, and refined the phrasing of several sentences to enhance the accuracy of the information presented.
Comments 8: The statement: “Researchers indicate that agonists targeting the MOR and KOR subtypes of opioid receptors are effective in pain alleviation.” should be reworded. All subtypes of opioid receptors have implications in analgesia.
Response 8: All subtypes of opioid receptors have implications for analgesia. We have revised the statement in the paper, which is shown in red font on line 72.
Comments 9: In the PZM21 section, the authors state that PZM21 is an analog of TRV130. This statement should be corrected, as PZM21 is not an analog of TRV130; rather it was identified through high-throughput screening.
Response 9: Unlike TRV130, the design strategy of PZM21 is based on the analysis of the solved MOR structure and is further developed through high-throughput screening. We have revised the statement in the paper, which is shown in the red font on lines 323-325.
Comments 10: There are various grammatical errors throughout the manuscript that need correcting. Some of these are minor, but some make it hard to determine what the authors are trying to convey.
Response 10: We thoroughly reviewed the entire text, corrected grammatical errors, and refined the phrasing of several sentences to enhance the accuracy of the information presented.
We would like to express our gratitude for your invaluable suggestions, which have significantly enhanced the completeness of our article. We eagerly anticipate your review and hope to receive your favorable feedback.
Reviewer 2 Report
Comments and Suggestions for Authors
In this review, Ju and colleagues focus on biased agonism that is displayed by clinical opiates. The authors cover the signaling pathways associated with mu opioid receptors including both G protein- and beta arrestin-mediated pathways. This topic is timely given the worldwide opioid epidemic and the research that is involved in trying to design safer medications to treat pain. The authors do a nice job of covering the topic. However, I have some major concerns that detract from this paper.
First, the section that covers the signaling by opioid receptors is not entirely correct (Lines 77 -84). As stated, this section gives the impression that Galpha subunits cause the block of Ca2+ channels and opening of K+ channels. It is the Gbeta/gamma subunits that bind to Ca2+ channels or K+ (GIRK) channels and block Ca2+ channels or open GIRK channels. This is somewhat presented in the Figure 1 legend. However, Figure 1 is also somewhat incorrect as it shows the arrow with a plus sign going to K+ channels and is an arrow that starts from the Galpha protein. What the authors should do is make sure that the arrow starting from Gbeta/gamma targets both the Ca2+ and K+ channels.
Second, the authors should consider asking a native English speaking person read the entire manuscripts. There are many instances of wrong grammar and it is a detraction for the reader.
Comments on the Quality of English LanguageThe quality of English has to be improved. There are grammatical errors throughout the manuscript.
Author Response
Response to Reviewer 2:
We would like to express our sincere gratitude for your valuable feedback, which can enhance the quality of our manuscript. According to your constructive suggestions, we have made revisions to the manuscript, as follows:
Comments 1: In this review, Ju and colleagues focus on biased agonism that is displayed by clinical opiates. The authors cover the signaling pathways associated with mu opioid receptors including both G protein- and beta arrestin-mediated pathways. This topic is timely given the worldwide opioid epidemic and the research that is involved in trying to design safer medications to treat pain. The authors do a nice job of covering the topic. However, I have some major concerns that detract from this paper.
First, the section that covers the signaling by opioid receptors is not entirely correct (Lines 77 -84). As stated, this section gives the impression that Galpha subunits cause the block of Ca2+ channels and opening of K+ channels. It is the Gbeta/gamma subunits that bind to Ca2+ channels or K+ (GIRK) channels and block Ca2+ channels or open GIRK channels. This is somewhat presented in the Figure 1 legend. However, Figure 1 is also somewhat incorrect as it shows the arrow with a plus sign going to K+ channels and is an arrow that starts from the Galpha protein. What the authors should do is make sure that the arrow starting from Gbeta/gamma targets both the Ca2+ and K+ channels.
Response 1: MOR is activated and coupled with G protein, which subsequently dissociates into α and βγ subunits. These two molecules inhibit adenylate cyclase (AC) activity, thereby preventing adenosine triphosphate (ATP) from producing cyclic adenosine monophosphate (cAMP). This process simultaneously leads to the inactivation of calcium channels (reducing Ca2+ influx) and the activation of potassium channels (increasing K+ efflux). Therefore, we modified the transmission part of MOR signaling and made corresponding adjustments to the figure, as indicated in Figure 1 and in red on lines 77-91.
Comments 2: Second, the authors should consider asking a native English speaking person read the entire manuscripts. There are many instances of wrong grammar and it is a detraction for the reader.
Response 2: We sincerely apologize for the numerous grammatical errors in the paper. We have thoroughly reviewed the content and refined the language to enhance clarity and precision in expression.
Thank you again for your valuable suggestions, which have made our article more complete. Looking forward to your review again and hoping to receive your appreciation.
Round 2
Reviewer 2 Report
Comments and Suggestions for Authors
In my previous critique, I informed the authors that they were mistaken on how they described G protein signaling. I specifically wrote that beta and gamma subunits block the Ca2+ channels and activate K+ (GIRK) channels. Unfortunately, the authors still are mistaken with this signaling and is clear that they do not understand this process. The authors should read a review article on how G protein beta/gamma subunits. In the current version, the authors for some unknown reason insist that G beta/gamma adenylyl cyclase activity. That does NOT happen. Until the authors get this correctly, this paper should not be published as it contains serious errors.
Comments on the Quality of English Language
The grammar and English is improved, but there are still several grammatical errors. It is NOT THE JOB of the reviewer to carefully check the entire manuscript and search for mistakes. Thus, the paper needs major revisions.
Author Response
Comments: In my previous critique, I informed the authors that they were mistaken on how they described G protein signaling. I specifically wrote that beta and gamma subunits block the Ca2+ channels and activate K+ (GIRK) channels. Unfortunately, the authors still are mistaken with this signaling and is clear that they do not understand this process. The authors should read a review article on how G protein beta/gamma subunits. In the current version, the authors for some unknown reason insist that G beta/gamma adenylyl cyclase activity. That does NOT happen. Until the authors get this correctly, this paper should not be published as it contains serious errors.
Response: Thank you very much for your comments. We apologize for the ambiguity regarding the section on G protein signal transduction, and we realize that the figure has not been modified appropriately. We have taken your suggestions into account, reviewed a publication on opioid receptor signaling in the journal Cell (Che T et al., 2023, Cell), and corrected the fundamental errors in the manuscript, as indicated in Figure 1 and highlighted in red on lines 95-101. Additionally, we carefully reviewed the manuscript again, corrected grammar errors, and improved sentence expression to enhance the accuracy of the presented information.
Round 3
Reviewer 2 Report
Comments and Suggestions for Authors
The authors have addressed my concerns.